# Enhancing Immunogenicity in Metastatic Melanoma: Adjuvant Therapies to Promote the Anti-Tumor Immune Response

**DOI:** 10.3390/biomedicines11082245

**Published:** 2023-08-10

**Authors:** Sandra Pelka, Chandan Guha

**Affiliations:** 1Department of Development and Molecular Biology, Albert Einstein College of Medicine, Bronx, NY 10461, USA; sandra.pelka@einsteinmed.edu; 2Department of Radiation Oncology, Albert Einstein College of Medicine, Bronx, NY 10461, USA; 3Department of Pathology, Albert Einstein College of Medicine, Bronx, NY 10461, USA; 4Department of Urology, Albert Einstein College of Medicine, Bronx, NY 10461, USA; 5Institute of Onco-Physics, Albert Einstein College of Medicine, Bronx, NY 10461, USA

**Keywords:** melanoma, immunogenic cell death, anti-tumor immune response, damage-associated molecular patterns, focused ultrasound

## Abstract

Advanced melanoma is an aggressive form of skin cancer characterized by low survival rates. Less than 50% of advanced melanoma patients respond to current therapies, and of those patients that do respond, many present with tumor recurrence due to resistance. The immunosuppressive tumor-immune microenvironment (TIME) remains a major obstacle in melanoma therapy. Adjuvant treatment modalities that enhance anti-tumor immune cell function are associated with improved patient response. One potential mechanism to stimulate the anti-tumor immune response is by inducing immunogenic cell death (ICD) in tumors. ICD leads to the release of damage-associated molecular patterns within the TIME, subsequently promoting antigen presentation and anti-tumor immunity. This review summarizes relevant concepts and mechanisms underlying ICD and introduces the potential of non-ablative low-intensity focused ultrasound (LOFU) as an immune-priming therapy that can be combined with ICD-inducing focal ablative therapies to promote an anti-melanoma immune response.

## 1. Introduction

Melanoma is a rare and aggressive form of skin cancer that is responsible for 75% of skin cancer-related deaths [1,2]. When diagnosed in advanced stages, melanoma is associated with a very poor prognosis with a patient survival rate as low as 30% [2]. Immune checkpoint inhibitors (CIs), such as anti-CTLA-4 and anti-PD-1 blocking antibodies (Table 1), target the inhibitory receptors on T-cells, including programmed cell death protein-1 (PD-1) and cytotoxic T-lymphocyte-associated protein-4 (CTLA-4), with the goal of enhancing anti-tumor T-cell function. These checkpoint inhibitors have significantly improved survival rates in patients with advanced melanoma. 

CI treatment efficacy has been shown to correlate with the number of tumor-infiltrating lymphocytes, indicating that an endogenous anti-tumor T-cell immunity is associated with increased response [15]. However, CI therapies are only effective in 10–40% of patients when used as monotherapies, and approximately 53% when combined as dual therapies [16]. Even in cases where CI therapy is effective initially, many patients proceed to present with tumor recurrence. Potential mechanisms underlying this loss of CI efficacy include the selection for tumor cells that are resistant to CI therapy, the upregulation of other inhibitory receptors, and the clearance of CI antibodies by tumor-associated macrophages [17,18]. Metastatic melanoma cells have also been shown to release exosomes, extracellular vesicles that are approximately 30 to 150 nm in size, with a high expression of PD-L1 which subsequently suppresses the anti-tumor immune response [19]. Conversely, the expression of PD-L1 by myeloid-derived immune cells such as conventional dendritic cells and macrophages within the tumor microenvironment is required for the generation of a strong response to CI therapies [20,21]. Additionally, CI therapies are associated with adverse events, some of which can be severe and which include autoimmune-related disorders such as vitiligo and colitis, and even death [22]. The overall incidence of immune-related adverse events (IRAE) in advanced melanoma patients is dependent on the checkpoint inhibitor therapy regimen: approximately 23% of patients treated with pembrolizumab monotherapy experience IRAE, and this incidence increases to 62% with ipilimumab monotherapy and 83% with combined CI therapies [23]. Unfortunately, there is a positive association between CI efficacy and number of adverse events [24]. Therefore, balancing the low response rate and the risk of adverse effects is vital when determining whether a patient should receive CI therapy.

Biomarkers are a potential tool to guide personalized treatment plans for melanoma patients. The programmed death-ligand 1 (PD-L1), expressed on tumor cells, binds to T-cell surface PD-1, thereby attenuating the anti-tumor T-cell response. Although PD-L1 expression, measured by immunohistochemistry, has been approved by the Food and Drug Administration (FDA) as a biomarker to predict responsiveness to immunotherapy in certain cancers (non-small cell lung cancer, gastric/gastroesophageal junction adenocarcinoma, triple-negative breast cancer, and others), the utility of this biomarker is still under investigation [25,26]. However, such biomarkers may play an important role in directing patient therapy in the future.

A major challenge in cancer immunotherapy is the immunosuppressive tumor-immune microenvironment (TIME). The generation of a robust anti-tumor response relies on dendritic cells (DCs), professional antigen-presenting cells which must first recognize tumor antigens and damage-associated molecular patterns (DAMP) within the TIME via pattern-recognition receptors, and then process the antigens and migrate to the draining lymph node, where they present tumor antigen peptides to both CD4+ and CD8+ T-cells [27]. However, within the TIME, immunosuppressive factors impede DC differentiation and activation, leading to systemic changes in DC populations and function that prevent efficient antigen presentation [28]. As a result, melanoma patients present with decreased numbers of peripheral and intra-tumoral DCs [29,30,31,32]. Additionally, DC subpopulations in melanoma patients favor an immature phenotype and lack co-stimulatory signals [30,31,33,34]. These immature DCs can lead to the establishment of T-cell anergy, in which T-cells, the critical effectors of the TIME, enter a dysfunctional, tolerant state with notably reduced effector cytokine secretion and proliferation [35,36,37]. In melanoma, intra-tumoral populations of T-cells often show anergic and exhausted phenotypes [38], and these populations are associated with increased melanoma persistence [34]. 

## 2. Immunogenic Cell Death (ICD)

Focal therapies that induce ICD in tumor cells have the potential to overcome immunosuppression in the TIME and to improve patient responses to current immunotherapies [39]. Tumor-cell ICD triggers a local, anti-tumor immune reaction (Figure 1) that can potentially prevent resistance to CI therapies [39,40]. Typically, tumor cells that are undergoing ICD experience endoplasmic reticular (ER) stress due to oxidative stress and/or the accumulation of misfolded proteins in the ER, with subsequent activation of the unfolded protein response (UPR) [39,40]. This leads to the translocation of calreticulin from the endoplasmic reticulum (ER) to the plasma membrane (ecto-CRT) [39,40,41]. Ecto-CRT promotes the phagocytosis of tumor cells by DCs [42]. Additionally, ecto-CRT also promotes natural killer cell-mediated tumor cell death [43]. Studies have shown that ecto-CRT is essential for the induction of ICD, since melanoma cells that are unable to translocate CRT cannot stimulate ICD and are subsequently unable to induce an anti-tumor immune response [41]. The cofactors required for the translocation of calreticulin are also involved in other cellular processes such as autophagy that have also been implicated in pro-immunogenic signaling within the TIME.

ICD, immunogenic cell death; ER, endoplasmic reticulum; UPR, unfolded protein response; PERK, protein kinase RNA-like ER kinase; ATF4, activating transcription factor 4; DAMP, damage-associated molecular pattern; HMGB1, high-mobility group box protein 1; ATP, adenosine triphosphate (Created with BioRender.com).

Autophagy, a lysosome-dependent process of cytosolic content degradation, plays important roles in both immune evasion [44,45,46] and tumor growth [47,48,49]. Recent studies have shown that inducing autophagy in melanoma cells can protect them from immediate apoptosis and that this delay in apoptosis is necessary for the generation of pro-immunogenic signaling and ICD markers [50]. Increased levels of autophagy in tumor cells prior to ICD is required for the secretion of adenosine triphosphate (ATP), which acts as a damage-associated molecular pattern (DAMP) and chemoattractant to enhance the migration of inflammatory monocytes and DCs to the tumor site. Cancer cell lines depleted of autophagy-specific machinery are not able to activate a tumor-specific immune response [51,52]. Although autophagy-deficient cells can undergo apoptosis and necrosis, they show a reduction in secreted ATP, which is required to enhance the recruitment of immune cells to the tumor site [51]. Increased levels of autophagy within tumors has been shown to be positively correlated with improved patient prognosis [52]. 

As tumor cells undergo ICD, the dying cells also release high-mobility group box protein 1 (HMGB1). HMGB1 is normally found within the nucleus, but as cells are damaged, it can translocate to the cytosol and be secreted into the extracellular space, where it acts as a DAMP that subsequently activates the immune response [53]. HMGB1 directly and indirectly interacts with a variety of receptors including toll-like receptor 4 (TLR4) and the receptor for advanced glycation end-products (RAGE) [48]. HMGB1 binding with TLR4 on antigen-presenting cells leads to the direct induction of an adaptive anti-tumor immune response, while its binding with RAGE induces autophagy [54], which may subsequently induce pro-immunogenic signaling mechanisms. Dying tumor cells have also been shown to release nucleic acids such as RNA that bind to toll-like receptor 3 (TLR3) and activate innate immune cells to secrete type-1 interferons (IFN-1s) [55,56]. IFN-1s have been implicated in the stimulation of both the innate and adaptive anti-tumor immune response and are currently being investigated as a potential target for novel therapies [56].

Enhancing ICD in melanoma patients may lead to a better patient prognosis. Ren et al. identified a group of three differentially expressed genes (GBP2, THBS4, and APOBEC3G) in melanoma patients that are related to ICD and can be utilized to predict patient responses to immunotherapies [57]. Further investigation has shown a positive correlation of tumor-cell ICD with improved patient prognosis [57]. Adjuvant therapies that induce ICD have already been shown to act synergistically with current checkpoint inhibitor therapies by activating DC antigen presentation and T-cell infiltration within the TIME [58]. Hence, focal ICD-inducing therapies may sensitize tumors to immunotherapies and may be effective in combination therapies that improve patient quality of life without adding the systemic adverse effects of systemic chemotherapy.

## 3. Targets of ICD

### 3.1. Transcriptional Activators and Epigenetic Targets

To better identify potential targets that can be utilized to promote an anti-tumor immune response, it is important to first understand the mechanisms underlying melanoma cell growth and development. Melanoma stems from the pigment-producing cells of the skin, called melanocytes. One of the transcription factors that plays an important role in melanocyte development, proliferation, and survival is the microphthalmia-associated transcription factor (MITF) [59]. High levels of MITF expression have been shown to favor a non-invasive, proliferative melanoma phenotype, while low levels of MITF expression favor an invasive, metastatic melanoma phenotype [59]. In melanoma patients, the signal transducer and activator of transcription 3 (STAT3), which is known to antagonize MITF expression [60], has been found to play a role in melanoma ICD. One study found that using stattic, a STAT3 inhibitor, in combination with the chemotherapeutic and known ICD-inducer doxorubicin, leads to increased ecto-CRT, increased release of HMGB1, and increased expression of 70-kilodalton heat shock proteins (HSP70) [61], suggesting that STAT3 is a potential target for the induction of melanoma ICD.

Another potential target to increase melanoma tumor immunogenicity is the transcriptional activator SOX10, which also regulates MITF [62,63]. SOX10 plays an important role in neural crest development and inhibits melanoma cell immunogenicity by inducing the expression of interferon regulatory factor 4 (IRF4), a negative regulator of interferon regulatory factor 1 (IRF1) transcription [64]. IRF1 is a tumor suppressor known to promote the activation of IFN-1 genes, another hallmark of ICD [65]. Since SOX10 has been shown to negatively correlate with PD-L1 expression on tumor cells, targeting SOX10 may not only enhance tumor cell ICD, but may also improve responses to anti-PD-1/-PD-L1 therapies by its direct effects on PD-L1 expression [64]. Therapeutics such as histone deacetylase inhibitors may be used to suppress SOX10 expression, thus leading to the induction of PD-L1 expression on melanoma cells and increased responsiveness to anti-PD-L1 therapies [64,66].

In addition to histone deacetylase inhibitors, which have been shown to improve melanoma cell responsiveness to therapy, other epigenetic modulations have also been shown to promote immunogenicity within the TIME. For example, a dual inhibitor of histone and DNA methyltransferases was shown to induce apoptosis in B16-F10 melanoma and subsequently resulted in increased activation of CD4+ and CD8+ T-cells within the TIME and tumor growth delay [67]. The chromatin modifier lysine-specific demethylase 1 (LSD1) is an example of an epigenetic target that has also been shown to inhibit anti-tumor immunity [68]. Sheng et al. showed that B16 melanoma tumors that were deficient in LSD1 had increased secretion of IFN-1s, leading to increased T-cell infiltration within the TIME [68]. Additionally, the LSD1-deficient melanoma cells had increased sensitivity to anti-PD-1 therapy [68], suggesting that epigenetic targets that promote ICD marker expression may be utilized to promote tumor immunogenicity and to improve responsiveness to CI therapy.

### 3.2. ICD Signaling Pathway Targets

ICD is associated with the accumulation of misfolded and unfolded proteins in the ER, resulting in the induction of ER stress and the UPR. The protein kinase RNA-like ER kinase (PERK) arm of the UPR plays a role in ICD [69,70]. PERK has downstream effects on both pro-apoptotic and pro-survival pathways. It also induces the translation of activating transcription factor 4 (ATF4), which plays a key role in the regulation of autophagy [71]. Conversely, the ablation of PERK signaling within melanoma cells induces paraptosis, a specific type of programmed cell death that is also associated with ER stress, increased secretion of IFN-1, and the migration of inflammatory DCs in the TIME [72]. 

Other types of cell death have also been associated with ICD and an increased susceptibility to CI therapy [73,74]. Ferroptosis, a form of iron-dependent, non-apoptotic cell death, leads to the accumulation of reactive oxygen species and is emerging as a promising target for adjuvant therapies [75]. Regulators of ferroptosis have shown promise as therapeutic targets. For example, calcium/calmodulin-dependent protein kinase kinase 2 (CAMKK2) inhibits ferroptosis [76,77], and its expression has been identified as a potential driver in CI therapy resistance [78]. This suggests that targeting CAMKK2 and its signaling pathways may subsequently enhance ferroptosis and immunogenicity within the TIME.

ICD can also be induced via innate immune system pathways. For example, retinoic acid-inducible gene I (RIG-I) is a cytosolic RNA sensor that, when stimulated, can induce ICD via the interferon-dependent apoptosis of melanoma cells [79,80,81]. RIG-I agonists, such as M8, can be used to activate the RIG-I pathway, subsequently activating the innate immune system by increasing DC expression of the co-stimulatory factors CD80 and CD86, and further enhancing DCs’ ability to process antigens and cross-present to CD8+ T-cells [79,80,81].

TLRs are an example of pattern-recognition receptors that recognize pathogen-associated molecular patterns and DAMPs. They also play a major role in initiating the innate immune response and, as a result, are an attractive target for increasing susceptibility to current CI therapies [82,83,84]. In addition to promoting an innate immune response within the TIME, TLR agonists, such as the TLR-2/3 agonist, L-pampo (LP) [85], and the TLR-7 agonist, Imiquimod [86], have been shown to induce tumor-cell ICD, as evidenced by enhanced ecto-CRT and the increased release of HMGB1 and ATP [85,86]. Both TLR agonists promote a CD8+ T-cell- and IFNγ-driven anti-tumor response and lead to increased CD8+ T-cell infiltration in B16-F10 tumors [86].

The direct targeting of molecules and mechanisms known to play a role in ICD (Table 2) may enhance the efficient induction of an anti-tumor adaptive immune response. However, depending on the immunogenicity of a tumor and its associated mutations, sometimes, multiple mechanisms may be needed to promote pro-immunogenic signaling within the TIME. Additional therapies that induce tumor-cell ICD by targeting alternative mechanisms (Table 3) may be used as adjuvants to enhance the endogenous anti-tumor response and to increase patient responses to current gold standard therapies.

## 4. Novel Pro-Immunogenic Adjuvant Therapies

### 4.1. Nanovesicles for Local Targeting of Chemical ICD-Inducers

Some chemotherapeutics, including doxorubicin and paclitaxel, promote anti-tumor immunity by promoting ICD [96]. Doxorubicin has also been shown to increase the expression of PD-L1 by tumor cells [87], which may promote T-cell exhaustion within the TIME [87]. Combination therapies with ICD inducers may enhance the responsiveness to anti-PD-L1 therapy. Celastrol, a member of the quinone methide family, may enhance the immune response via multiple pathways, including the induction of tumor cell autophagy, suppression of PD-L1 expression by melanoma cells, and induction of ICD [88]. 

Other ICD-inducers under investigation for use in melanoma include chromomycins A5-8, antibiotics that induce tumor cell ICD, autophagy, apoptosis, and ecto-CRT [89]; and polyphenols, which promote the release of calcium from the ER and subsequently disrupt the mitochondrial membrane potential, leading to ICD [90]. Another ICD-inducer under investigation is peptide RT53, which induced ICD and complete tumor regression in a B16-F10 mouse model [91]. 

However, one of the challenges with chemical inducers of ICD is the potential for severe side effects, including cardiotoxicity and neurotoxicity [97]. Targeted therapies that allow the local delivery of ICD-inducers directly to the tumor site are a major area of research. Nanovesicles are an attractive therapeutic tool since they can be conjugated with targeting ligands and can be modulated to retain their cargo until they reach the tumor site. Acidic pH or near-infrared (NIR) light can then trigger the intra-tumoral release of ICD-inducers from nanovesicles [98], thereby mitigating the systemic, off-target adverse effects. Studies have shown that the nanovesicular delivery of ICD-inducers leads to increased activation of DCs as compared to when ICD-inducers are delivered without nanovesicular targeting [97]. Nanovesicles have the added benefit of allowing the loading and targeted delivery of combinatorial treatments. One group designed a nanovesicular platform that is loaded with multiple ICD-inducers in addition to an indoleamine 2,3-dioxygenase (IDO) inhibitor [99]. This combination therapy takes advantage of another mechanistic target of anti-tumor therapies: metabolism. The metabolic profiles of tumor cells influence their ability to activate the immune response and affect immune cell functions [47,100,101,102,103,104,105]. For example, tumor cells expressing the tryptophan-catabolizing enzyme IDO are known to lead to T-cell dysfunction [36,106,107,108]. IDO, therefore, is an attractive target that may complement ICD’s effects on the adaptive immune response. Certain compounds, such as the tryptanthrin derivative CY-1-4, both inhibit IDO and promote ICD [109] and have been investigated in melanoma models as potential therapeutics. 

### 4.2. Oncolytic Viruses as Cancer Vaccines

Oncolytic viruses are of interest due to their ability to induce and secrete DAMPs and tumor antigens, while reprogramming and immune-activating the stromal cells in the TIME [110]. These viruses infect and lyse cancer cells and have already been used in human patients with promising results. Talimogene laherparepvec (T-VEC), an oncolytic herpes simplex virus HSV-1 that encodes the granulocyte-macrophage colony-stimulating factor, has already been FDA-approved for the treatment of melanoma [111]. Although the mechanism of action has not yet been fully characterized, T-VEC has been shown to promote ICD by increasing ecto-CRT and the release of HMGB1 and ATP [111]. Researchers have also found that the response to T-VEC appears to be inversely correlated with the expression of the stimulator of interferon genes (STING). When melanoma cells are deficient in the STING, there is increased tumor cell death in response to T-VEC which leads to increased CD8+ T-cell recruitment and infiltration to the tumor site [92,93]. Further modifying current oncolytic virus therapies to encode additional fusion proteins, such as the highly fusogenic form of the envelope glycoprotein of gibbon ape leukemia virus (GALV-GP-R-), can further enhance their immunogenic effect by promoting abscopal effects and the memory T-cell response [94]. Targeting ligands can also be used to further enhance the pro-immunogenic effects of cancer vaccines. One such potential target is CD47, a membrane protein expressed by tumor cells that prevents them from being recognized and phagocytosed by macrophages [112,113]. An adenovirus-based tumor vaccine loaded with a CD47-targeting nanobody fused with the IgG2a Fc protein was found to induce anti-tumoral immunity and promote tumor regression and overall long-term survival in mice [114].

Other oncolytic viruses have also been evaluated as promising immunomodulatory tools to create a “hot” pro-immunogenic tumor microenvironment [115,116,117]. Although not all of these act by inducing ICD, they are being investigated as complementary therapies that work synergistically with ICD-inducers. For example, FixVac (BNT111), an intravenously administered liposomal RNA vaccine (Clinical Trial NCT02410733), was shown to enhance anti-tumor immunity when used in combination with ICD-inducers [95].

### 4.3. Carbon Ion Radiotherapy

Carbon ion radiotherapy (CIRT), also called heavy ion therapy, is an emerging form of radiation therapy that uses carbon ion/nuclei beam radiation to primarily treat solid tumors. One of the advantages of CIRT is its increased precision in targeting the tumor site, thus mitigating off-target adverse effects due to the characteristic energy distribution of particles, known as the “Bragg Peak” [118]. CIRT also has a greater relative biological effectiveness compared to other forms of radiation therapy since the damage caused by the carbon ions is highly focused in the DNA and thus is able to efficiently overwhelm the cellular stress response [118]. As of January 2023, there are currently 14 centers worldwide that provide patients with CIRT [119]. Primarily used in the treatment of solid tumors such as melanoma, CIRT promotes tumor immunogenicity by enhancing the translocation of calreticulin to the plasma membrane, enhancing the release of HMGB1 and ATP, and promoting an IFN1 response, all major hallmarks of ICD [120]. CIRT has currently been investigated in melanoma in combination with anti-PD-1 therapy and was shown to enhance CD4+ and CD8+ tumor infiltration [120]. It has also been shown to decrease populations of immunosuppressive cells, such as myeloid-derived suppressor cells, within the TIME, further supporting a pro-immunogenic effect [121]. In clinical trials, CIRT has also been shown to improve patients’ progression-free survival, suggesting that it is a promising adjuvant treatment to current immunotherapies [122].

### 4.4. Photodynamic and Photothermal Therapy

Phototherapy relies on the application of a photosensitizer, typically a chemical compound or nanoparticle, that is excited by a light source to generate either local hyperthermia (photothermal therapy) or reactive oxygen species (photodynamic therapy) at a desired target location. These stressors subsequently affect cellular homeostasis and activate cellular stress responses and ICD [123]. Recently, various photosynthetic drugs and nanoparticle drug delivery systems for the treatment of melanoma were reviewed [124]. Known ICD-inducers such as doxorubicin have already been paired with photothermal agents such as indocyanine green as a mechanism to further sensitize tumor cells to ICD induction [125]. Current research is focused on identifying novel photosensitizers that can be administered for phototherapy, often in the form of multi-component nanovesicles or polymeric networks in order to optimize targeting to the tumor site [126,127]. For example, Konda et al. identified two NIR-absorbing ruthenium (II) complexes, ML19B01 and ML19B02, that promote tumor cell ICD upon activation by NIR [128]. Graphene oxide nanosheets are also an example of a photosensitizer, which, upon exposure to NIR light, induce local hyperthermia at the tumor site, leading to an increased expression of heat shock proteins and subsequent ICD [129].

Studies to enhance photothermal therapy effects have also investigated other photosensitizing mechanisms. Optical droplet vaporization is a strategy that was previously used to improve ultrasound imaging, in which perfluorocarbon droplets are superheated and vaporized to a gas phase [130]. This procedure was also found to cause direct cellular damage and activate cellular stress responses and, as a result, is now also applied with photothermal therapy [130].

Photodynamic therapy of B16 melanoma leads to enhanced ecto-CRT and an increased release of DAMPs, including HMGB1 and IFN1 [131]. Phototherapy also leads to additional downstream effects on the anti-tumor immune response, including the enhanced phagocytic activity of monocytes and enhanced cross-presentation to CD8+ T-cells, thus increasing CD8+ T-cell activation and proliferation [131]. The overall goal of phototherapy is to precisely target the tumor site in order to mitigate off-target adverse effects. Additionally, the immunogenic effects seen with phototherapy suggest that this therapy holds promise as a tool that can remodel the TIME and induce systemic, tumor-specific immunity.

### 4.5. Focused Ultrasound

Therapeutic focused ultrasounds (FUSs) can be used as a non-invasive adjuvant therapy to induce thermal and/or mechanical stress focally at the tumor site. A piezoelectric transducer is used to convert an electrical signal into ultrasonic waves, characterized by periods of compression and periods of rarefaction, measured by the peak positive pressure and peak negative pressure, respectively. High-intensity FUS (HIFU), where the delivered total energy intensity is typically greater than 1000 Watts/cm^2^, has already been applied in clinical trials for various solid cancers. HIFU primarily acts by inducing thermal coagulative necrosis, which is associated with non-specific tissue destruction and inflammation, and the induction of cellular stress responses [132,133,134,135,136]. FUS has been shown to promote a T-cell-mediated anti-tumor response, characterized by decreased tumor growth and proliferation [137]; increased infiltration of activated, tumor-specific effector CD4+ and CD8+ T-cells; and increased infiltration of inflammatory macrophages [138,139]. In a B16-F10 melanoma mouse model, the combination therapy of FUS with an anti-CD40 agonistic antibody activated DCs by upregulating costimulatory molecules (ex: CD80, CD86) and MHC class II expression [139]. This subsequently induced melanoma-specific T-cell immunity and also suppressed the growth of a secondary, untreated tumor, suggesting that FUS may promote an abscopal effect by inducing systemic anti-tumoral immunity [139]. 

In addition to promoting immunogenicity at the tumor site, FUS has also been combined with nanobubbles to enhance drug delivery. The nanobubbles contain inert gas, which, upon application of FUS, begin to oscillate, expanding and collapsing, generating a phenomenon called the cavitation effect [140]. The delivery of ultrasound pulses at the tumor site and subsequent oscillations cause increased permeability of the affected cells’ membranes, increasing both the number and the size of membrane pores and facilitating drug delivery and efficacy [141,142]. When given in combination with CI therapy, the adjuvant FUS + nanobubble treatment enhances the CD8+ T-cell anti-tumor response and generates long-term memory, suggesting that it may likewise contribute to an abscopal effect [140]. This parallels the effects of another treatment modality called electrochemotherapy, in which an electrical field is applied to destabilize the cell membrane and improve drug diffusion into cells [143]. FUS is one of few non-invasive adjuvant therapies that can be used to enhance the T-cell-mediated anti-tumor response both by inducing hyperthermia and increasing membrane permeability.

### 4.6. Sonodynamic Therapy

FUS has also gained interest within the context of sonodynamic therapy, which refers to a combination therapy using FUS with an intra-tumoral injection of a sensitizing agent, such as a nickel ferrite/carbon nanocomposite [144]. This leads to both increased ICD as well as increased membrane permeability for synergistic, pro-immunogenic effects. Sonodynamic therapy (SDT) has been used to target brain metastasis in melanoma and can promote tumor cell damage and apoptosis via the generation of reactive oxygen species and cavitation [145]. When applied in vivo, SDT induced approximately 60% necrosis in a B16-F10 tumor model [144]. When FUS was combined with doxorubicin-releasing microspheres, it was found to extend progression-free survival in a melanoma model, further emphasizing the promising potential of these dual, pro-immunogenic therapies [146].

### 4.7. Magnetic Hyperthermia

Magnetic hyperthermia is a form of nanovesicle-mediated, localized hyperthermia that induces tumor cell death. Magnetic hyperthermia delivered via Fe_3_O_4_ nanoparticles has been shown to induce necrotic ICD and has been used both in preclinical mouse studies and in preliminary clinical trials in patients with advanced metastatic melanoma [147,148]. In mouse melanoma models, magnetic hyperthermia combined with CI therapy led to enhanced tumor regression compared to CI therapy alone. Furthermore, its combination with radiotherapy increased the expression of chemoattractants and the activation of TLR pathways in the TIME, indicating that magnetic hyperthermia can enhance the immune activation in TIME [147,148]. In a preliminary clinical trial of magnetic hyperthermia currently underway at the Sapporo Medical University in Japan (Clinical Trial Research Protocol No. 18-67), of the four patients treated so far, one patient showed complete tumor regression, and one patient showed a partial anti-tumor response [148]. While further studies are still needed, magnetic hyperthermia remains a prospective tool to locally induce tumor ICD and enhance immunogenicity within the TIME.

**Table 3 biomedicines-11-02245-t003:** Treatment modalities that promote ICD and their mechanisms of action.

ICD Therapy Platform	Mechanism	References
Nanovesicles	Platform that utilizes targeting ligands to deliver cargo to tumor site; mitigate systemic adverse effects; acidic pH or NIR can trigger cargo release; allow co-loading of cargo with other sensitizers/ICD-inducers; leads to increased DC activation	[97,99]
Carbon ion radiotherapy (CIRT)	Carbon ion (^12^C^6+^) beam radiation is directed at the tumor site; enhances CD4+ and CD8+ tumor infiltration; decreases myeloid-derived suppressor cell infiltration of the TIME	[120,121]
Photodynamic/photothermal therapy	Photosensitizer is excited by a light source to generate local hyperthermia (photothermal therapy) or ROS (photodynamic therapy) at the tumor site	[123]
Focused ultrasound (FUS)	Leads to non-specific tissue destruction and inflammation (HIFU) or to mild hyperthermia with increased heat shock protein expression and induction of cellular stress responses (LOFU); combination with nanovesicles and microbubbles leads to increased cell membrane permeability and improved drug delivery	[132,133,134,135,136,140]
Sonodynamic therapy	Combination therapy of FUS with a sensitizing agent; generates ROS and promotes tumor cell damage, apoptosis, and necrosis	[145]
Magnetic hyperthermia	Nanovesicle-mediated, localized hyperthermia induces necrotic ICD; increases expression of chemoattractant and TLR pathway markers	[147,148]
Nanosecond pulsed electric fields	Electric pulses enhance membrane permeability and trigger cellular stress responses (autophagy, necrosis, and apoptosis)	[149,150]
Plasma-derived oxidants	Increases presence of ROS/RNS	[151,152]

### 4.8. Nanosecond Pulsed Electric Fields

Nanosecond pulsed electric fields (nsPEFs) are another potential adjuvant therapy that can be applied to ablate melanoma tumors. However, studies have shown a high variation in tumor response, likely due to differences in the electrical field parameters among research groups [149,150,153]. While some studies suggest that nsPEFs increase immunogenicity and the induction of cellular processes such as autophagy [149,150], recent work by Rossi et al. suggests that necrosis may be the primary mechanism of cell death resulting from nsPEFs [153]. Overall, nsPEF treatment appears to enhance membrane permeability and trigger cellular stress responses, including autophagy, necrosis, and apoptosis [149,150]. Additional studies are needed to better identify the effect of nsPEF on melanoma tumor models and to determine the efficacy of nsPEF as a tool to enhance TIME immunogenicity.

### 4.9. Plasma-Derived Pro-Oxidant Treatment Modalities

Partially ionized gas or plasma has been used for biological and medical applications for several decades. For example, a hot argon plasma coagulator is used to cauterize telangiectasia to stop rectal bleeding from radiation proctitis. In contrast, cold physical plasma generates a multitude of reactive oxygen and nitrogen species (ROS/RNS) that can cause mitochondrial and ER stress and ICD in tumor cells, such as melanoma. When chemotherapeutics such as doxorubicin, epirubicin, and oxaliplatin are combined with cold plasma-derived oxidants, they upregulate organic cationic transporter SLC22A16 that facilitates cytotoxic drug uptake, increases the secretion of ICD-associated DAMPs such as ATP, and enhances the tumoricidal effects of chemotherapeutics [154]. Non-thermal plasma is therefore under investigation as an inducer of ICD and potential adjuvant therapy, primarily due to the presence of ICD stressors, specifically ROS/RNS [151,152]. 

## 5. Limitations of Current Therapies

Many of the above therapeutic modalities have been shown to promote the adaptive immune response by enhancing the pro-immunogenic ICD. This is especially important within the TIME, which is typically characterized by immunosuppressive cell phenotypes. For example, a major mechanism underlying T-cell tolerance in the TIME involves immature DCs, which have a low expression of co-stimulatory molecules and can lead to the establishment of T-cell anergy [34]. Anergic or tolerant T-cells are defined by transcriptome reprogramming which leads to an unresponsive state with notably reduced cytokine secretion and proliferation [35,37,155]. Similarly, exhausted T-cells, generated in the TIME due to repeated antigen exposure, also secrete lower amounts of effector cytokines and are unable to induce a robust and long-term anti-tumor response [37,38]. In melanoma, intra-tumoral populations of T-cells often show anergic and exhausted phenotypes, and these populations have been associated with increased melanoma persistence [34,36,38].

While ICD-inducing, ablative therapies have been shown to promote tumor immunogenicity, they are still primarily used as focal therapies for the local ablation of tumors. The true test of their efficacy will be in their ability to reprogram the immunosuppressive features of the TIME and to sustain an immune-activating TIME that will facilitate tumor antigen presentation and the cross-priming of both CD4+ and CD8+ T cells. A sustained anti-tumoral immune response that can achieve control of systemic metastases after the induction of ICD in tumor cells by focal ablative therapies has not been commonly observed [156]. To date, there is also limited knowledge on whether ICD-inducing therapies are able to reverse or prevent T-cell anergy. Thus, further work is still required to determine whether these ICD-inducing therapeutic modalities may adequately establish long-term anti-tumor function.

## 6. Low-Energy Focused Ultrasound (LOFU) for Immune-Priming of ICD-Inducing Therapies

LOFUs are performed by applying a lower-intensity ultrasound to tissue, with a maximal intensity limit of 800 Watts/cm^2^. Unlike HIFU, which is already being applied as a form of ablative cancer therapy for certain solid malignancies in the United States, LOFU is non-invasive and primarily induces a thermal and mechanical cellular stress response, activating the pre-apoptotic phase of ICD and increasing the expression of heat shock proteins [157,158]. When applied in a B16 murine melanoma tumor model, LOFU, combined with an ablative radiation therapy, was found to lead to long-term, T-cell-dependent control of melanoma tumor growth [157]. The most promising aspect of LOFU treatment is its potential to serve as a source of immune priming. In a B16 melanoma model, LOFU treatment, compared to no treatment, led to a decreased transcriptional expression of multiple anergy-associated genes, including three E3 ubiquitin ligase genes named GRAIL, Cbl-b, and Itch within CD4+ T-cells isolated from tumor-draining lymph nodes [157]. All these E3 ubiquitin ligases were previously shown to be transcriptionally upregulated when T-cell anergy was induced via sustained calcium-calcineurin signaling [159], thus suggesting that LOFU treatment, upon decreasing their transcription, may promote a reversal of anergy in tumor-draining CD4+ T cells. These findings set LOFU aside from other current ICD-inducing therapy modalities, which have not yet been shown to reverse T-cell tolerance. Although the mechanisms by which LOFU may reverse T-cell anergy have not yet been fully characterized, future work in this area is vital to identify targets to overcome the immunosuppressive TIME and to prevent future tumor growth/recurrence.

## 7. Conclusions

The current gold standard therapies for melanoma are not sufficient to overcome the poor survival rates seen in advanced-stage melanoma patients. Considering the low response rate, the potential for tumor recurrence following therapy, and the immune-related adverse effects associated with CI therapies, there remains a need for adjuvant therapies to further improve patient survival and quality of life. CD4+ and CD8+ T-cells are considered the primary effectors of the anti-tumor response, but in the setting of melanoma, they often enter an anergic and exhausted state. An attractive approach to overcome the tolerant TIME is to promote an endogenous immune response by introducing DAMPs and activating pro-immunogenic signaling pathways. ICD has therefore been garnering more attention as a vital mechanism to promote the anti-tumor response. Therapies that induce ICD stressors directly at the tumor site have the potential to simultaneously enhance the endogenous anti-tumor immune response while mitigating off-target adverse effects. Of the discussed ICD-inducing treatment modalities, LOFU especially shows promise in its ability to reprogram the TIME and promote long-term, pro-immunogenic anti-tumor effects due to its potential to reverse T-cell anergy. Furthermore, LOFU can be combined as an immune-priming therapy prior to traditional, focal ablative therapies. Future research to better understand how these various therapies induce ICD is needed to continue enhancing tumor immunogenicity and to provide novel targets for future treatment strategies.

## Figures and Tables

**Figure 1 biomedicines-11-02245-f001:**
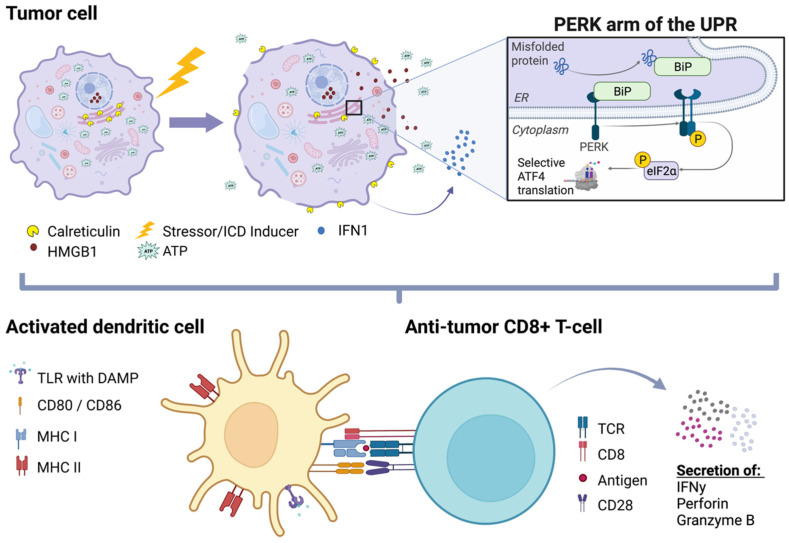
**Schematic representation of ICD and its effect on the anti-tumor immune response.** After exposure to a stressor or ICD-inducer, tumor cells experience ER stress and subsequently upregulate the UPR. One of the three arms of the UPR which involves PERK has been mechanistically implicated in the induction of ICD and is also associated with the regulation of autophagy through ATF4. Upregulation of the UPR may lead to various forms of ICD, characterized by the translocation of calreticulin (CRT) from the ER to the plasma membrane and the release of DAMPs including HMGB1 and ATP. Immature DCs phagocytose soluble tumor antigens and/or dying tumor cells with ecto-CRT “eat me” signals. Tumor cells undergoing ICD also release type-1 interferons (IFN1), essential drivers of anti-tumor immunity. The released DAMPs interact with TLRs on DCs within the tumor microenvironment, leading to the maturation and activation of DCs with increased expression of MHC-II and the co-stimulatory molecules CD80 and CD86. Mature DCs migrate to the draining lymph node and present tumor antigens via both class II and class I MHC pathways to prime and activate CD4+ and CD8+ T-cells, respectively. CD8+ T-cells are the primary effectors of the anti-tumor adaptive immune response and are able to directly kill tumor cells using perforin and granzyme B.

**Table 1 biomedicines-11-02245-t001:** Current FDA-approved systemic melanoma therapies.

Therapy	Generic Name (Brand Name)	References
Checkpoint inhibitor therapies	Nivolumab and Relatlimab Combination Therapy (Opdualag)	[3]
	Ipilimumab (Yervoy)	[4,5]
	Pembrolizumab (Keytruda)	[6]
	Nivolumab (Opdivo)	[7]
BRAF/MEK inhibitors	Binimetinib (Mektovi) + Encorafenib (Braftovi)	[8]
	Vemurafenib (Zelboraf) + Cobimetinib (Cotellic)	[9]
	Dabrafenib (Tafinlar) + Trametinib (Mekinist)	[10,11]
Cytotoxic chemotherapy	Dacarbazine	[12]
Other immunotherapies	Interleukin-2 (Aldesleukin, Proleukin)	[13]
	Recombinant Interferon Alfa-2b (Intron A)	[14]

**Table 2 biomedicines-11-02245-t002:** Targeted ICD-inducers and their mechanisms of action.

ICD-Inducer	Examples	Mechanism	References
Chemical/small molecule inducers	Chemotherapeutics (i.e., doxorubicin and paclitaxel)	Increase tumor cell expression of PD-L1	[87]
	Celastrol (quinone methide family)	Induces tumor cell autophagy; decreases tumor cell expression of PD-L1	[88]
	Chromomycins A5-8 (antibiotics)	Induces tumor cell apoptosis	[89]
	Polyphenols	Induces PERK arm of the UPR	[90]
	RT53 (peptide)	Promotes B16-F10 tumor regression	[91]
Oncolytic viruses	Talimogene laherparepvec (T-VEC)	Oncolytic herpes simplex virus HSV-1 that infects tumor cells and encodes granulocyte-macrophage colony stimulating factor; Enhances CD8+ T-cell recruitment via STING-mediated pathway; Modulation to express GALV-GP-R^-^ leads to enhanced ICD and abscopal effect	[92,93,94]
	FixVac (BNT111)	Liposomal RNA vaccine (Clinical Trial NCT02410733) that enhances anti-tumor immune response when used with other ICD-inducers	[95]

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
