# Peer review of "Enhancing Immunogenicity in Metastatic Melanoma: Adjuvant Therapies to Promote the Anti-Tumor Immune Response"

_biomedicines, 2023, doi:10.3390/biomedicines11082245_

Round 1

Reviewer 1 Report

In this review Sandra Pelka and Chandan Guha summarizes concepts and mechanisms underlying tumour cell immunogenic cell death (ICD) and discusses emerging methods for utilizing immunogenic cell death to promote an anti-melanoma endogenous immune response. The authors place significant emphasis on LOFU in the abstract, however, only a small part of the review discusses this. I suggest rewriting the abstract, focusing on the main message of the Review. There are only 2 papers cited about the effect of LOFU.

References: The names of the journals is not consistnet: e.g.

1. Siegel, R.L.; Miller, K.D.; Wagle, N.S.; Jemal, A. Cancer statistics, 2023. CA: A Cancer Journal for Clinicians 2023, 73, 17-48, 527 doi:https://doi.org/10.3322/caac.21763. 528

2. Davis, L.E.; Shalin, S.C.; Tackett, A.J. Current state of melanoma diagnosis and treatment. Cancer biology & therapy 2019, 20, 1366-529 1379.

DOI is missing from most of the references; The name of journals: each word should started with capital letter

Author Response

Reviewer 1 Comment:

In this review Sandra Pelka and Chandan Guha summarizes concepts and mechanisms underlying tumour cell immunogenic cell death (ICD) and discusses emerging methods for utilizing immunogenic cell death to promote an anti-melanoma endogenous immune response. The authors place significant emphasis on LOFU in the abstract, however, only a small part of the review discusses this. I suggest rewriting the abstract, focusing on the main message of the Review. There are only 2 papers cited about the effect of LOFU.

Response:

Based on comments from Reviewer 1 and Reviewer 3, we have made significant revisions to our abstract. Our new abstract is below:

“Advanced melanoma is an aggressive form of skin cancer characterized by low survival rates. Less than 50% of advanced melanoma patients respond to current therapies, and of those patients that do respond, many present with tumor recurrence due to resistance. The immunosuppressive tumor immune microenvironment (TIME) remains a major obstacle in melanoma therapy. Adju-vant treatment modalities that enhance anti-tumor immune cell function are associated with im-proved patient response. One potential mechanism to stimulate the anti-tumor immune response is by inducing immunogenic cell death (ICD) in tumors. ICD leads to the release of dam-age-associated molecular patterns within the TIME, subsequently promoting antigen presentation and anti-tumor immunity. This review summarizes relevant concepts and mechanisms underly-ing ICD and introduces the potential of non-ablative low intensity focused ultrasound (LOFU) as an immune-priming therapy that can be combined with ICD-inducing focal ablative therapies to promote an anti-melanoma immune response.”

Reviewer 1 Comment: 

References: The names of the journals is not consistent: e.g. 

  1. Siegel, R.L.; Miller, K.D.; Wagle, N.S.; Jemal, A. Cancer statistics, 2023. CA: A Cancer Journal for Clinicians 2023, 73, 17-48, 527 doi:https://doi.org/10.3322/caac.21763. 528 
  2. Davis, L.E.; Shalin, S.C.; Tackett, A.J. Current state of melanoma diagnosis and treatment. Cancer biology & therapy 2019, 20, 1366-529 1379. 

DOI is missing from most of the references; The name of journals: each word should started with capital letter 

Response:

Thank you for bringing this to our attention. We have updated our references so that the journal names are consistently in title case and have added DOIs to all relevant references as well.

Reviewer 2 Report

I believe that the review submitted for publication by Sandra Pelka and Chandan Guha is of wide interest not only to clinicians involved in anticancer therapies but also to melanoma investigators. The review appears well-structured and comprehensive of current topics on the treatment of malignant melanoma.

Author Response

Reviewer 2 Comment:

I believe that the review submitted for publication by Sandra Pelka and Chandan Guha is of wide interest not only to clinicians involved in anticancer therapies but also to melanoma investigators. The review appears well-structured and comprehensive of current topics on the treatment of malignant melanoma.

Response:

Thank you for your positive comments. Based on comments from Reviewers 1 and 3, we have made revisions to our manuscript so that it may read more clearly and have also updated our references to include DOIs where necessary.

Reviewer 3 Report

Odd words.

Phrases misplaced. 

Some paragraphs with difficult comprehension.

Author Response

Reviewer 3 Comment:

Overall, the review is well structured with a good rationale. It starts by introducing melanoma and its features, and then the authors describe the strategies applied. They began with the ICD and essential aspects, then presented different strategies to elicit ICD. Finally, they show the limitations and give an idea of what they consider the best approach with the LOFU. Each section has good information and proper references.

Nevertheless, the reviewer consider that there are some aspects that need to be improved:

  1. Abstract: It has all the information but the English is poor. Consider revising it.

Response:

We have made significant revisions to our abstract so that the wording may be more clear. Our new abstract is below:

“Advanced melanoma is an aggressive form of skin cancer characterized by low survival rates. Less than 50% of advanced melanoma patients respond to current therapies, and of those patients that do respond, many present with tumor recurrence due to resistance. The immunosuppressive tumor immune microenvironment (TIME) remains a major obstacle in melanoma therapy. Adju-vant treatment modalities that enhance anti-tumor immune cell function are associated with im-proved patient response. One potential mechanism to stimulate the anti-tumor immune response is by inducing immunogenic cell death (ICD) in tumors. ICD leads to the release of dam-age-associated molecular patterns within the TIME, subsequently promoting antigen presentation and anti-tumor immunity. This review summarizes relevant concepts and mechanisms underly-ing ICD and introduces the potential of non-ablative low intensity focused ultrasound (LOFU) as an immune-priming therapy that can be combined with ICD-inducing focal ablative therapies to promote an anti-melanoma immune response.”

Reviewer 3 Comment:

  1. All the text: some words are odd. Some statements are misplaced. Consider revising it.

Response:

We have reviewed the manuscript in its entirety and have revised and rephrased sentences as needed so that it may be more clear. These sentences have been highlighted in the resubmitted manuscript.

Reviewer 3 Comment:

  1. A table with the current drugs used in chemotherapy and immunotherapy must be added! Melanoma was the first cancer type being treated with anti-CTLA4 (ipilimumab) and also with anti-PD-1 (nivolumab, pembrolizumab …). Now, there are more, so it would be nice to have that in a table.

Response:

As recommended, we have included a table of current melanoma drugs, now Table 1. We chose to focus on current FDA-approved therapies, and have included both generic and brand names of these select therapies.

Reviewer 3 Comment:

  1. Figure 1 is missing…

Response:

We apologize for this and have re-inserted Figure 1 within the manuscript text.

Reviewer 3 Comment:

  1. Table 1 and 2 – put the reference in a different column.

Response:

We have created a new column in each of these tables and have moved the references as suggested. Please note that these are now Table 2 and Table 3 respectively. Additionally, we have adjusted the wording within our tables so that they are more concise and clear.

Reviewer 3 Comment:

  1. Do you know of any clinical trial with immunotherapy and ICD inducers? If yes, I think that a table with that information would be beneficial.

Response:

Thank you for your suggestion. We have included information regarding current applications of ICD inducers to the best of our knowledge within the text of the manuscript.

Reviewer 3 Comment:

  1. In the conclusion, the authors can put a statement with the benefits of LOFU, since it is highlighted in the manuscript.

Response:

As recommended, we have revised the conclusion to include some final statements regarding LOFU therapy and its potential benefits.

Reviewer 3 Comment:

  1. References: if you add the doi in one reference, the author should had in all.

Response:

Thank you for bringing this to our attention. We have updated our references and have added DOIs to all relevant references.

Reviewer 3 Comment:

Specific:  Line 50 – 30 to 150 nm

Response:

Thank you for bringing this to our attention. We have formatted this within our text accordingly (now Line 46)

Reviewer 3 Comment:

Line 54 – This is a strong statement. Are they frequent? If not, that information must be there; otherwise, the authors are transmitting the idea that CI are very toxic.

Line 56 – sometimes it means that the treatment is being effective. Mention if these effects are mild, or severe, and enumerate a few.

Response:

Thank you for your comments. We have included statistics regarding the frequency of these adverse effects with different treatment modalities, and have adjusted our wording. Please see Lines 50-57.

Reviewer 3 Comment:

Line 77 - reference

Response:

Thank you for bringing this to our attention. We have included a reference for this statement. Please see Line 88.

Reviewer 3 Comment:

Line 241 – ICD, Line 518 – ICD

Response:

Thank you for bringing this to our attention. We have updated our text to use the abbreviations throughout as you recommended. We have additionally reviewed and updated our text so that all abbreviations (ex: UPR, ecto-CRT, PERK, ICD, DC, etc) are used consistently. Instances of where abbreviations were added to replace text have been highlighted throughout the manuscript.

Reviewer 3 Comment:

Line 512 – Strong statement! With anti-CTLA4, the adverse events were difficult to manage but with anti-PD-1 is different.

Response:

We appreciate your comments. We have adjusted the wording for this statement to address your concerns. Please see Lines 486-488.

Round 2

Reviewer 1 Report

Thank you for the answers and for ther corrected version of your manuscript.

Reviewer 3 Report

The authors replied to all my queries.